# Selection of Reference Genes in *Siraitia siamensis* and Expression Patterns of Genes Involved in Mogrosides Biosynthesis

**DOI:** 10.3390/plants13172449

**Published:** 2024-09-02

**Authors:** Wenqiang Chen, Xiaodong Lin, Yan Wang, Detian Mu, Changming Mo, Huaxue Huang, Huan Zhao, Zuliang Luo, Dai Liu, Iain W. Wilson, Deyou Qiu, Qi Tang

**Affiliations:** 1Yuelushan Lab, College of Horticulture, Hunan Agricultural University, Changsha 410128, China; 15958699677@163.com (W.C.); 13452762262@139.com (X.L.); wang17873769091@sina.com (Y.W.); mudetian12580@163.com (D.M.); 2Guangxi Crop Genetic Improvement and Biotechnology Lab, Guangxi Academy of Agricultural Sciences, Nanning 530007, China; 3Hunan Huacheng Biotech, Inc., High-Tech Zone, Changsha 410205, China; river@huachengbio.com (H.H.); 258253167@163.com (D.L.); 4School of Traditional Chinese Medicine, Capital Medical University, Beijing 100069, China; 53522722@163.com; 5Institute of Medicinal Plant Development, Chinese Academy of Medical Sciences, Peking Union Medical College, Beijing 100193, China; zlluo@implad.ac.cn; 6CSIRO Agriculture and Food, Canberra, ACT 2601, Australia; iain.wilson@csiro.au; 7State Key Laboratory of Tree Genetics and Breeding, Research Institute of Forestry, Chinese Academy of Forestry, Beijing 100091, China; qiudy@caf.ac.cn

**Keywords:** *Siraitia siamensis*, reference genes, RT-qPCR, mogrosides, synthesis pathways

## Abstract

*Siraitia siamensis* is a traditional Chinese medicinal herb. In this study, using *S. siamensis* cultivated in vitro, twelve candidate reference genes under various treatments were analyzed for their expression stability by using algorithms such as GeNorm, NormFinder, BestKeeper, Delta CT, and RefFinder. The selected reference genes were then used to characterize the gene expression of *cucurbitadienol synthase*, which is a rate-limiting enzyme for mogroside biosynthesis. The results showed that *CDC6* and *NCBP2* expression was the most stable across all treatments and are the best reference genes under the tested conditions. Utilizing the validated reference genes, we analyzed the expression profiles of genes related to the synthesis pathway of mogroside in *S. siamensis* in response to a range of abiotic stresses. The findings of this study provide clear standards for gene expression normalization in Siraitia plants and exploring the rationale behind differential gene expression related to mogroside synthesis pathways.

## 1. Introduction

*Siraitia siamensis* is a major perennial vine of Cucurbitaceae family with significant financial value [1]. It has been used for many years by traditional Chinese medicine for the treatment of congested lungs, common colds, and laryngitis [2,3]. Siraitia species include *Siraitia grosvenorii* and *S. siamensis*, whose main bioactive compounds are mogrosides, a type of triterpenoid sweetener. The fruit extracts can be used as sugar-free health food and beverage supplements and sweeteners [4] because of the presence of naturally low-calorie mogroside V. It can be used as a sugar substitute for people with diabetes, as it does not raise blood sugar levels [5]. *S. siamensis* is rich in siamenoside I, being about 560 times as sweet as sucrose and approximately 1.4 times as sweet as aspartame [6]. Due to its excellent disease and insect resistance properties, *S. siamensis* has increasingly become a focus of research. The pharmacologically important secondary metabolites of this species have been the focus of research interest [7].

Secondary metabolites in traditional Chinese medicinal plants have been studied more intensively in recent years; however, medicinal plants generally have common problems such as slow growth cycles and complex extraction processes [8] that hamper their investigation. Plant tissue culture technology offers an excellent option for these species for large-scale production of healthy plants and tissues with increased amounts of secondary metabolites from medicinal plants, independent of their natural growing season. Currently, there are no reports on the in vitro tissue culture of *S. siamensis*; therefore, the development of an in vitro culture and propagation protocol for this species is urgently needed for research purposes. By analyzing and modifying the methods of in vitro tissue culture for *S. grosvenorii* [9], the results were obtained through orthogonal experiments.

Modern plant physiology research has shown that exogenous hormones (e.g., SA, MeJA, EtH) and temperature extremes (low and high) force plants to alter the biosynthesis of secondary metabolites [10,11,12]. Methyl jasmonate interacts with pathway genes and affects metabolite accumulation by regulating genes associated with the biosynthesis of pentacyclic triterpenoids in *Ocimum basilicum* [13]. Salicylic acid treatment has validated the importance of *PgCYP736B* in *Panax ginseng* signaling to produce the secondary metabolite ginsenoside [14]. High temperature induces the opening of Ca^2+^ channels and the increase in ganoderic acid content during biosynthesis in *Ganoderma lucidum* [15]. A low-temperature treatment elevates an increase in H_2_S signaling molecules, leading to cucurbitacin C content in cucumber and affecting plant adaptation to cold stress [16]. Nevertheless, there is still a big gap in the study of the effect of exogenous phytohormones on the biosynthesis of mogrosides in *S. siamensis*. However, studies on the effects of exogenous phytohormones on the biosynthesis of mogrosides in *S. siamensis* are still relatively lacking. Thus, to further identify genes related to this pathway, it is essential to study their expression patterns under various biotic stresses.

The synthesis of these metabolites can be effectively manipulated, and the underlying molecular mechanisms need to be identified. Mogrosides are tetracyclic triterpenoids and based on isoprene as the backbone [17]. Isopentenyl diphosphate (IPP) and dimethylallyl pyrophosphate (DMAPP) form the common precursors of all isoprenoids. This precursor can be synthesized via two different pathways: The mevalonate (MVA) pathway in the cytoplasm and the 2-C-methl-D-erythritol-4 (MEP) pathway in the plasmid [18]. The MVA pathway takes acetyl CoA as the starting substrate and generates IPP through six steps of condensation reaction, and plays the leading role in the biosynthesis of triterpenoid saponins [19]. IPP and DMAPP were condensed under *GPS* to form geranylpyrophosphate (GPP, C10). A second IPP unit was added to the GPP to generate farnesyl pyrophosphate (FPP, C15) under the catalysis of *FPS* [20]. The two FPPs are then condensed by *SQS* to squalene (C30). This is then epoxidized by *SQE* to 2,3-oxidosqualene (C30) and the continued oxidation to form 2,3,22,23-dioxidosqualene [21]. *Cucurbitadienol synthase (CS)* is the first rate-limiting enzyme in the downstream synthesis stage of mogroside saponins, which guides 2,3-oxidoxysqualene to complete the cyclization process to generate a triterpenoid backbone 24,25 epoxycucurbitadienol [22]. 24,25 epoxycucurbitadienol opens epoxy under the action of *Epoxide hydrolase* enzyme to form 24,25 dihydroxy-cucurbitadienol. *Cytochrome P450 monooxygenase* (*CYP450*) is responsible for the hydroxylation at C3, C11, C24, and C25 required to produce mogrol in the mogroside biosynthesis pathway. Among these, the CYP87D18 enzyme is a multifunctional enzyme that oxidizes the C-11 position of cucurbitadienol to form mogrol and 11-O-mogrol [23]. Subsequent glycosylation modification with mogrol as the aglycone resulted in the formation of mogroside V [24]. *S. siamensis* has genes related to the biosynthesis of mogrosides, so their exploitation at the molecular level has considerable research potential (Figure 1).

The utilization of quantitative real-time PCR (RT-qPCR) in the context of gene expression analysis may prove to be a valuable tool for the elucidation of secondary metabolite metabolism in a particular tissue under varying conditions [25]. RT-qPCR is now extensively used in molecular biology studies and is a reliable method for determining gene expression levels. RT-qPCR is known for its accuracy, convenience, speed, and sensitivity [26]. The results from this type of analysis are known to be prone to interference by multiple factors, so RT-qPCR requires a reference gene as an internally generated control to standardize the expression level of the target gene. Preferred reference genes must be stably expressed in different biological or abiotic conditions, in different tissues, and are not affected by endogenous or exogenous factors [27]. *S. grosvenorii*, another medicinally important plant species, belongs to the same genus as *S. siamensis* and has many reported reference genes that have greatly accelerated the analysis of its genes expression profiles. Wei used *UBQC* as the reference gene to analyze the expression of *dehydroascorbate reductase* (*DHAR*) gene in *S. grosvenorii* [28]. Zhao used *GAPDH* gene as the reference gene of RT-qPCR to analyze the expression level of *Cucurbitane synthase (CS)* gene at different periods [29]. And *EF1α* [30], *RPL13* [31], *TIP41* [32], *tubB* [33], *GAPDH* [34], *tubA* [35], and *NCBP2* [36] have been widely used as reference genes for a multitude of plant species. Currently, there are no reports on the selection and validation of reference genes for *S. siamensis*. To ensure reliable RT-qPCR studies in *S. siamensis*, it is necessary to identify reference genes.

A panel of 12 reference genes was selected for the analysis based on a review of the pertinent scientific literature and the genome data: *RPL13*, *CDC6*, *TIP41*, *tubB2*, *tubB3*, *GAPDH*, *tubA*, *EF1α*, *NCBP2*, *UBQC*, *PP2A*, and *PCACO.* The stability of the expression of these genes were assessed under a variety of abiotic pressures such as SA, MeJA, ETH, and high and low temperature. Five algorithms (GeNorm, NormFinder, BestKeeper, ∆CT, and RefFinder) were used for the systematic screening of optimal reference genes [37,38,39,40,41]. To validate the reliability of the determined these genes, an expression spectrum was created for each of the *CS* genes. Finally, the expression pattern of mogrosides synthesis pathway genes under different treatments was analyzed using the new reference genes [2,24]. This study determined the optimal appropriate reference genes for RT-qPCR in the context of different abiotic stresses, thereby establishing the basis for molecular gene expression studies on *S. siamensis* in the future.

## 2. Results

### 2.1. Plant Regeneration

Using basic Murashige and Skoog’s (MS) tissue culture conditions, without the use of any phytohormones, segments of *S. siamensis* were sufficient to establish and propagate under in vitro conditions. In the case of the asexual propagation of *S. siamensis*, there are no studies on the induction of shoot segments from explants into plants. Though all combinations tested (Materials and Methods) were successful in inducing explants to form plants, faster proliferating plants were produced on a C5 medium as compared to other combinations (Figure 2). The leaves produced on C5 showed a dark green color and were healthier and exhibited superior vigor, thus warranting further asexual propagation (Appendix A). The plants underwent a process of passaging and culturing at 15-day intervals.

### 2.2. Reference Gene Selection, Amplification Specificity and PCR Efficiency Evaluation

Based on previous scientific studies on related plant species, twelve candidate normalization reference genes (*RPL13*, *CDC6*, *TIP41*, *tubB2*, *tubB3*, *GAPDH*, *tubA*, *EF1α*, *NCBP2*, *UBQC*, *PP2A*, and *PCACO)* were selected for further analysis of their stability profiles under a range of stress conditions known to affect secondary metabolite formation. Table 1 lists the abbreviations and names of screened reference genes with their primer sequences, amplification efficiencies (E), and correlation coefficients (*R*^2^). The specificity of the internal reference genes was determined via RT-qPCR melting curve analysis. The melting curves of all genes displayed an individual peak, as shown in Appendix A, and the reproducibility of the RT-qPCR amplicon curves was satisfactory. The amplification rates for the 12 reference genes were between 98.8% and 102.6%, with the amplification efficiencies of *GAPDH* and *RPL-13* being the lowest and the highest, respectively (Table 1 and Appendix A). The correlation coefficients of the twelve reference genes were > 0.99, which are suitable for experimentation.

### 2.3. Expression Abundance Analysis of Candidate Reference Genes

The cycling threshold (Ct) is used to establish the frequency of cycling needed for a generated fluorescent signature to achieve a detectable level. The expression abundance of the twelve candidate genes was measured by calculating the RT-qPCR Ct value. The lower the Ct value, the higher the abundance of gene expression, provided that the amplicons exhibit comparable response efficiencies. As illustrated in Figure 3, the Ct values of the majority of genes exhibited a distribution between 17 and 25. The gene exhibiting the highest level of expression was PCACO (17.35 ± 0.99), while the gene exhibiting the lowest level of expression was CDC6 (23.78 ± 1.07). In addition, the length of the box plots is indicative of the stability of the data. Shorter box plots indicate that the data are concentrated in a narrower range, whereas longer box plots indicate that the data are more dispersed with significant differences. *PP2A* (21.26 ± 4.39) had the greatest variation in expression levels, whereas *GAPDH* (19.61 ± 0.50) had a smaller range of variability. Nevertheless, given the intricacy of their external environment and the necessity to guarantee the stability of the internal reference genes, we analyze their expression using a variety of algorithms.

### 2.4. Expression Stability Analysis of Candidate Reference Genes

When selecting the most stable reference genes, it is necessary to analyze their expression under abiotic stress conditions to determine which genes to include in the expression profiling analysis. Five different software algorithms (GeNorm (https://genorm.cmgg.be/, accessed on 4 April 2024), NormFinder (https://www.moma.dk/normfindersoftware/, accessed on 5 April 2024), Bestkeeper (https://www.gene-quantification.com/bestkeeper.html, accessed on 4 April 2024), Delta C and RefFinder (http://blooge.cn/RefFinder/, accessed on 5 April 2024) were used to assess the stability of the twelve candidate reference genes in *S. siamensis* leaves after abiotic stress treatments by calculating expression Ct values. The experimental data were totaled into five groups, MeJA, EtH, SA, high-temperature (High Tem), and low-temperature groups (Low Tem) (all samples); all candidate genes within these groups were subjected to joint analysis.

#### 2.4.1. GeNorm Analysis

Evaluation using the GeNorm software involves converting the original Ct values of the resulting genes to relative expression levels before determining the generated M values. GeNorm is capable of identifying the optimal number by calculating paired variation (Vn/n + 1). If Vn and Vn + 1 fall below 0.15, the most optimal choice of reference gene is n. In this study, the pairwise variance results Vn calculated with GeNorm under all five treatments were less than 0.15 (Figure 4), which suggests that it is advisable to normalize Ct values using two internal reference genes.

The M value reflects the gene expression stability. When the M value is lower, the gene expression stability is higher. The stability of the internal reference genes varied among the five various treatments, as shown in Figure 5. CDC6 and RPL13 were the strongest stable genes in High Tem and EtH treatments. Whereas, in the Low Tem, MeJA, and SA treatment, the highest stability reference gene was *TIP41*. Thus, the stability of gene expression varied significantly between treatments. Further research is needed to determine the most appropriate reference gene.

#### 2.4.2. NormFinder Analysis

NormFinder is a statistically based method intended to assess the stabilities of gene expression. Its analytical framework considers both intra- and intergroup variation. It also calculates the M-values of the twelve internal reference genes for the original Ct values, and M values are in a negative correlation with the stability of expression. The lower the M value, the higher the stationarity, and the more appropriate for the normalization process. Under Low Tem treatment, *NCBP2* (0.264) and *UBQC* (0.229) had the highest stability, and *tubB3* (0.753) had the lowest. Whereas *NCBP2* (0.181) and *tubB3* (0.306) had the highest stability, *GAPDH* (0.97) had the lowest stability under High Tem treatment. In the context of the MeJA, EtH, and SA treatment subsets, it has been observed that *NCBP2* and *CDC6* were the most stable with the highest scores, while *PP2A* had the lowest stable score (Figure 6 and Appendix A). In summary, *CDC6* and *NCBP2* were observed to be the most stable and had the highest scores among the five different treatments.

#### 2.4.3. BestKeeper Analysis

The BestKeeper algorithm is a direct tool for the evaluation of Ct values generated by RT-qPCR, offering a direct estimation of the standard deviation (SD) and the coefficient of variation (CV) associated with these values. This assessment is important for determining the stability of potential reference gene expression. The SD of 1 is used as the critical value, indicating that if the SD exceeds 1, the data are unstable and unsuitable for normalization. The CV value represents the degree of variation in gene expression across different treatment groups. A smaller CV value indicates greater stability in gene expression. There were different results under each of the five different treatments. Overall, *CDC6* gene had the best performance and was at the top in almost every treatment. *EF1α*, *NCBP2*, *TIP41*, and *tubB2* genes were relatively stable and performed favorably in some treatments (Table 2). Genes with SD > 1 (like *PCACO*) indicate low expression stability across treatment conditions and, as a consequence, are unsuitable for use in the process of normalization.

#### 2.4.4. Delta Ct Analysis

In order to assess the stability of the potential reference genes, the Delta Ct algorithm utilizes the SD value. It is noteworthy that a negative correlation exists between the SD value and stability (Figure 7 and Appendix A), with the gene with the smallest SD value being the most stable. The results showed that in the Low Tem treatment group, the most stable genes were *NCBP2* (0.51) and *UBQC* (0.51), and the least stable was *tubB3* (0.86). In the High Tem treatment group, the two most stable genes were *NCBP2* (0.65) and *tubB3* (0.68), in that order, while the least stable was *GAPDH* (1.09). *CDC6* and *NCBP2* were the most stabilizing genes in MeJA, SA, and EtH treatments, whereas *PP2A* was the most unstable. The outcome of this approach is very comparable to the NormFinder algorithm results. In general, *CDC6* and *NCBP2* were the two most suitable endogenous genes for normalization.

#### 2.4.5. RefFinder Analysis

RefFinder analytics is an overall stability evaluation approach that assigns appropriate values the outcome of individual reference genes in each algorithm and then calculates the gene average of the algorithm stability value weights across all algorithms to produce a composite ranking. It avoids the one-lopsidedness of individual algorithms and can analyze the expression stability comprehensively. The smaller the stability value, the better the gene stability. As shown in Figure 8 and Appendix A, *CDC6* and *NCBP2* are the most stable in all treatments except the two temperature treatments, which has some differences with those of GeNorm, NormFinder, Bestkeeper, and Delta Ct algorithms. *UBQC* and *EF1α* have better performance in the temperature treatments. Among the statistical methods utilized, *PP2A* ranks at the bottom in each algorithm.

In general, *CDC6* and *NCBP2* were the two most suitable endogenous genes for normalization. In general analysis, among the five algorithms, only these two genes appeared more times in the front of the ranking (Figure 8), indicating that the *CDC6* and *NCBP2* candidate references are more stable. The best reference genes in terms of the coefficient of variation were two, and the best internal genes were *CDC6* and *NCBP2* genes.

### 2.5. Validation of Reference Genes by the Key Gene Cucurbitadienol Synthase

To validate the reliable reference genes, the *SsCS* gene was normalized by using any one of the three most stable genes in each treatment and using a combination of stable reference genes or one of the least stable genes. The *SsCS gene* is a key gene in the mogrosides synthesis pathways, which is a rate-limiting step, so its expression can strongly affect production. MeJA and temperature are known to alter the external environment, leading to changes in *SsCS* expression.

As shown in Figure 9, when the three most stable genes were used individually or in combination as reference genes for normalization, the relative expression pattern of *SsCS* showed similar trends. However, when an unstable gene was used for relative quantification, the relative expression levels of *SsCS* showed significant fluctuations. For example, under Eth treatment from 0 to 48 h, when stable genes (*NCBP2*, *CDC6*, *UBQC* and their combinations) were used as reference genes, the expression level of *SsCS* was the highest at 24 h and lowest at 12 h, with an overall trend of first decreasing then increasing. In contrast, when the least stable gene (*PP2A*) was used, the expression level of *SsCS* was high at 3 h and 24 h, with a different overall trend (Figure 9A). It is evident that using unstable reference genes for gene expression analysis in *S. siamensis* can lead to unreliable results. Additionally, in the SA, MeJA, high-temperature, and low-temperature treatment groups, *CDC6* and *NCBP2* were stable reference genes, and the expression levels and trends in *SsCS* under different groups pointed to similar conclusions (Figure 9B–E).

### 2.6. Different Expression Patterns of Mogrosides Synthesis Pathways under Different Treatments

In order to further enrich the study of mogrosides synthesis pathways, a total of fourteen genes in *S. siamensis* were obtained by homology comparison analysis based on the synthesis pathway-related genes for the NCBI database [2,11]. *CDC6* and *NCBP2* were used as the reference gene to identify and validate mogrosides synthesis pathways genes whose expression alters with the application of MeJA, EtH, SA, and heat and cold stress (Table 1 and Appendix A). RT-qPCR analysis was performed on leaves from tissue culture seedlings treated with MeJA, SA, and EtH at 0, 3, 6, 12, 24, 48 h and on leaves subjected to high- and low-temperature treatments at 0, 6, 12, 24, 36, and 48 h (Figure 10).

Among the EtH treatments (Figure 10A), the fourteen genes were *SsAACT*, *SsHMGS*, *SsHMGR*, *SsIPI*, *SsFPS*, *SsSQS*, *SsSQE*, *SsCS*, and *SsCYP*; all show a generally up-down-up trend. *SsMK*, *SsPMK*, *SsMVD*, *SsGPS*, and *SsEPH* all showed an overall trend of first increasing to 3 h, then decreasing, and finally a general trend in consistency to expression found at 0 h. This expression pattern exhibits a significant effect of EtH on the genes of the mogrosides synthesis pathway, and this effect had both upward and downward effects. The expression patterns analysis under MeJA treatments (Figure 10B), consisting of *SsAACT*, *SsMK*, *SsIPI*, *SsGPS*, *SsMVD*, *SsEPH*, *SsSQS*, and *SsPMK* genes, mostly showed an overall increasing trend that significantly increased and peaked at 3 h. The *SsHMGS*, *SsHMGR*, *SsFPS*, *SsSQE*, *SsCYP*, *and SsCS* genes showed a downward trend and then an overall upward trend in expression. This expression pattern illustrates that the effect of MeJA on the mogrosides synthesis pathway genes in *S. siamensis* is more similar to that of EtH and that there is an increase and a decrease in this effect. Under SA treatment, all genes analyzed peaked at 3 h, followed by a drastic decline, and the plants did not return to baseline (0 h-like) even by 48 h, indicating that SA has a drastic inhibitory effect on the growth of *S. siamensis*, which also have a major impact on the mogrosides synthesis pathway genes. (Figure 10C).

Under the Low Tem treatment, the genes, with the exception of *MK*, appeared to increase and then decrease by 36 h, indicating that the low-temperature treatment had a greater impact on gene expression and may therefore significantly affect the mogrosides synthesis pathway in *S. siamensis* (Figure 10D). After the High Tem treatment, the *SsAACT*, *SsIPI*, *SsGPS*, *SsSQE*, and *SsPMK* genes showed an increase at 3 h, but then went to near base line levels. *SsHMGS*, *SsHMGR*, *SsFPS*, *SsMVD*, *SsEPH*, *SsCYP*, *SsSQS*, and *SsCS* genes showed a decreasing trend in the High Tem treatment (Figure 10E), which suppressed the expression of these genes. If these genes were not affected by the high-temperature treatment, there is some scientific justification for using high temperature to screen related advantageous varieties.

## 3. Discussion

Secondary metabolites of medicinal plants are very important and are usually the main active ingredients that are closely related to the expression of their synthesis pathways [2]. There is a significant impact on the synthesis of secondary metabolites under different treatments, and they are associated with the expression profiles of genes related to the synthesis of the pathway [42]. The main secondary metabolite in *S. siamensis* is the mogrosides, which is a novel sweetener that does not cause elevated blood glucose and weight gain [43]. Current research on the *S. siamensis* focuses on species collection and molecular markers, but molecular research is currently lacking, due in part to lack good reference genes that are needed in order to normalize expression data so as to deal with the differences between samples related to the RNA amount and quality. Some previous research has shown that the levels of expression of many common reference genes differ in various species, tissues, and under various laboratory conditions in the same genus. He et al., selected *NCBP2* and *TIP41* as one of the best reference genes in *Peucedanum praeruptorum* [44]. Mu et al. showed that SAM could be used as the best reference gene after treating the leaves of *Uncaria rhynchophylla* via MeJA and EtH [45]. Zhang et al. found different optimal endosperm genes in different treatments when treating leaves of *Gelsemium elegans* with different hormones [46]. Zhou et al. found that they had different reference genes in different periods and different parts of *Evodia rutaecarpa* [47]. Other reference genes have been used extensively as references for many plants.

There are no reports in *S. siamensis* on the identification of genes with stable expression profiles suitable for RT-qPCR normalization under stress treatments. In this study, we have selected twelve reference genes which were based on gene homologues used in other plants. When screening for different treatments with respect to the effects of the reference gene, a set of relevant conditions were analyzed. Heat and cold stress is a major threat to plant productivities and has an impact on the distribution of plants and crop yields, especially when it happens in the growth stage [48]. MeJA is considered to be one of the most common hormones used in the treatment of plants to produce a response that will have a greater impact on plant secondary metabolites [49]. EtH is unstable when dissolved in water and produces ethylene at the time of application, which can produce a stress response in the plant and affect the gene expression process [50]. SA is a compound that has a drastic effect on plants, and if the plant can tolerate SA treatment, it is often more responsive to adversity [51]. Therefore, the selection of reference genes needs to be performed under different conditions, which are likely to be used to study secondary metabolites in *S. siamensis*.

After successfully screening these reference genes, the expression of these genes in *S. siamensis* can be known from the overall distribution plot of Ct value at 0 h, and through the Genorm algorithm analysis, there is the need to convert the Ct values into M values to generate the relative expression level. The analysis of the variance value of Vn/Vn +1 is less than 0.15; it can be seen that the optimal number of combinations of internal reference genes is two, and the most unstable gene is the *tubB* gene. In the NormFinder algorithm, which is performed to assess the stability by calculating M-values within and between groups, *NCBP2* and *CDC6* performed better overall, and *UBQC* and tubB3 genes were more stable on individual treatments. From the BestKeeper algorithm, it is possible to directly calculate the raw Ct values to obtain the SD and CV values for evaluation, and *CDC6* performed better overall, The relatively stable genes of *EF1α*, *NCBP2*, *TIP41*, and *tubB2* genes were more stable in individual treatments. In the Delta Ct algorithm, the raw Ct values can be directly calculated to obtain the SD for evaluation, and *CDC6* was found to perform better overall. In general, *CDC6* and *NCBP2* were the two most suitable reference genes for normalization. The conclusions obtained from the algorithms vary due to different computational criteria and data preprocessing, but the differences in the Bestkeeper results were the most striking.

This experiment was carried out on plant seedling grown in tissue culture, as *S. siamensis* has not yet been cultivated, and so only a scarce number of wild varieties are available, and these *S. siamensis* varieties are classified as endangered [52]. The main active ingredient in *S. siamensis* is the mogrosides, and there are fourteen steps in the synthesis pathway. Genes associated with this pathway, especially the *CS* gene, were chosen for validation because it is the first rate-limiting enzyme for the synthesis of cucurbit alkane-type triterpene skeleton, and it is also the only cyclooxygenase in this pathway, and it catalyzes the generation of cucurbitdienol from the substrate, 2,3-oxidized squalene [21]. From the results, it can be seen that based on the gene expression pattern, after MeJA and EtH treatments, there were both increases and decreases in gene expression related to mogrosides synthesis, which is similar to the results of Xu’s study in the susceptible varieties of peach [53]. SA and High Tem treatments were inhibitory for the whole pathway, and if the treatment was to be continued in duration, it could cause irreversible damage to the plant, which was observed in Bharani’s peanut study [50]. After Low Tem treatment, the genes related to mogrosides synthesis were significantly affected for a short period of time, but the overall expression gradually returned to the original untreated plant levels.

The identification of stable reference genes, even under a range of abiotic stresses in our study, now makes it possible to carry out gene expression-related experiments such as transcriptomics analysis, gene expression regulation, and the study of the gene editing effects on the activation and inhibition of gene expression in *S. siamensis*.

## 4. Materials and Methods

### 4.1. Explant Material Acquisition and Preparation of Plant Tissue Culture

*S. siamensis* was collected from Guangxi Academy of Agricultural Sciences in Nanning, China. First, the explants were rinsed with distilled water containing two drops of detergent for 10 min. Subsequently, the explants were subjected to a 2 min rinse in 70% ethanol (*v*/*v*), followed by a 2 min immersion in a 0.1% mercuric chloride (*v*/*v*) (HgCl_2_) (SINOPHARM, Shanghai, China) solution. The explants were then rinsed a further three times in autoclaved distilled water and transferred to Murashige and Skoog’s basal medium (Colaber Corporation, Beijing, China) under a laminar flow hood for the initiation process. Adjust the pH of the culture medium to 5.7 ± 0.2, solidify with 8 g/L agar, and sterilize in a vessel at 121 °C for 20 min. Cultures inoculated were maintained at 24/20 ± 1 °C (except for treatments involving heat and cold), a photoperiod of 16/8 h (light/dark), and a relative humidity of 60~70%.

### 4.2. Explant Nutritive Tissue Induction

Surface sterilized explants meristem was utilized for nutritive tissue induction. Different combinations of NAA (Colaber Corporation, Beijing, China), IBA, and activated charcoal were utilized to screen for ideal concentrations and compositions of nutrient tissues to be induced. *S. siamensis* samples were chosen based on consistent growth momentum, the stem cut into segments with 1 stem node and 2 leaves under sterile conditions for experimental materials, and then inserted directly into the different culture medium, with 5 bottles/treatment, and repeated 3 times (Table 3). The cultivation conditions are 26 °C, light intensity 2000 lx, and duration 10 h/d.

### 4.3. Stress and Elicitor Treatment

Diverse inducers were given to *S. siamensis* explants with 1/2 MS as the basic medium. In total, 18 plants with identical growth were divided into 6 groups. The treatments included the following: ethylene, (ETH, 100 mM), methyl jasmonate (MeJA, 100 mM), heat stress (42 °C), cold stress (4 °C), and salicylic acid (SA, 100 mM). In hormonal treatments, plants were placed in 100 mM MeJA, 100 mM EtH, and 100 mM SA for 0, 3, 6, 12, 24, and 48 h. MeJA, EtH, and SA are custom-made solutions purchased from Colaber Corporation, (Beijing, China). For cold and heat stresses, the seedlings were kept in the room at 4 and 42 °C for 0, 6, 12, 24, 36, and 48 h, depending on the conditions. The control group was only a watering treatment group. All groups had three bio-replicates. The cut leaves washed with water, dried, and then quick-frozen in liquid nitrogen and preserved at −80 °C.

### 4.4. Total RNA Extraction and cDNA Synthesis

Rapidly freeze 50–100 mg of *S. siamensis* leaf samples in liquid nitrogen and then grind to a fine powder using a mortar and pestle. Total RNA was obtained via the trizol method. Purity and concentration of extracted RNA were determined by UV spectrophotometer Micro Drop (BIO-DL, Shanghai, China), and RNA integrity was measured by 1% agarose gel electrophoresis. The extracted total RNA (1000 ng) was subjected to reverse transcription reaction with EvoM-MLV RT Mix kit and gDNA Clean (Accurate, Changsha, China) and 20 μL volume according to the instructions to generate the first strand cDNA. The cDNA samples ultimately used were diluted 10-fold with Rnase-free water. The samples obtained above were stored to 20 °C for subsequent experimental use.

### 4.5. Candidate Genes Selection and Primer Design

To determine potential reference genes, HMM3.0 was used to filter the *S. siamensis* genome database (https://dataview.ncbi.nlm.nih.gov/object/SRR22947134, accessed on 2 January 2023), and a total of reference genes were identified, including *RPL-13*, *CDC6*, *TIP41*, *tubB2*, *tubB3*, *GAPDH*, *tubA*, *EF1α*, *NCBP2*, *UBQC*, *PP2A,* and *PCACO*. (Appendix A). Fourteen target genes related to the mogroside synthesis pathway were included: *AACT*, *HMGS*, *HMGR*, *MK*, *PMK*, *MVD*, *IPI*, *GPS*, *FPS*, *SQS*, *SQE*, *CS*, *CYP450*, and *EPH*. Twelve reference genes and fourteen signal genes with coding sequences were provided by the *S. siamensis* genome database. RT-qPCR primers were both designed by Beacon designer 7 (Appendix A) and synthesized at Shanghai Biotechnology Co. (Shanghai, China). The melting point between the forward and reverse primers was lower than 1 °C, the GC content was 40~55%, the amplicons were 90~200 bp in length, and the amplification efficiency should be between 90 and 120%. All primers’ specificity was determined by the method of melting curve.

### 4.6. RT-qPCR Conditions

The overall RT-qPCR reaction system consisted of 10 μL SYBR Green Premix Pro Taq HS qPCR Kit (Rox Plus) (Accurate Biotechnology, Changsha, China), 0.4 μL of F/R primers (10 mM), 50 ng of cDNA template, and 0.4 μL of ROX Reference Dye II and ddH2O for a total volume of 20 μL. The RT-qPCR reaction process was as follows: 95 °C, 1 min; 95 °C, 15 s, 60 °C, 30 s, and 40 cycles. Melting curves are established the presence of non-specific amplification and primer dimerization (ABI7300, Applied Biosystems, Waltham, MA, USA). The melting curve analysis consisted of temperature readings from 65 to 95 at 0.5 regular intervals and the results of the experiments were analyzed. Three replicates were used for sample tested.

Comparative changes in gene expression were calculated using the comparable 2^−ΔΔCT^ method [54]. In addition, a 5-fold dilution series (5^0^, 5^−1^, 5^−2^, 5^−3^, and 5^−4^) of template cDNA (~100 ng) was used for all samples. Using these cDNA templates, correlation coefficients (*R*^2^) were calculated from standard curves. The amplification efficiency (E) of the primer can be determined from the slope value using the equation E = 10^−1÷k^.

### 4.7. Analysis of Candidate Reference Gene Expression Stability

First, different treatments 0 h of template cDNA (~100 ng) were used for all samples. Box plots of the cycle threshold (Ct) distribution of the reference genes in the control group were plotted using GraphPad Prism 9 software. Three replicates were used for sample tested. Four statistical algorithms, GeNorm, Normfinder, BestKeeper, and Delta Ct, were used to assess the stability of reference genes in all experimental treatments. The data were analyzed using the mean Ct values of the three biological replicates. The results were presented as mean ± standard error (SEM). For GeNorm and Normfinder analyses, the original Ct values were transformed as relative quantitative values (Q-values) with the formula Q = 2^−∆CT^. [55]. For the GeNorm algorithm, the Q-value gives the stability of the expression (M) and the variability of the pair (V). Lower values of M are associated with higher stability. The V value determines the optimal number of reference genes, and when the Vn/n + 1 < is 0.15, n is the optimal value. The NormFinder algorithm also obtains M-values, but the stability is assessed primarily based on intra- and inter-sample variation in reference genes [38]. The BestKeeper algorithm requires the original Ct value, after which it utilizes the SD and CV as standards for steady expression of reference genes. In essence, a lower value denotes greater stability [39]. The ΔCT method is employed to assess the reliability of gene expression, with the mean SD of each gene being calculated as a metric. Finally, the RefFinder website (http://blooge.cn/RefFinder/ (accessed on 5 April 2024) was used to select the most appropriate reference genes by combining the above four algorithms to obtain the overall evaluation ranking [40,41].

### 4.8. Validation of Reference Genes

To validate the reference genes, we chose cucurbitdienol synthase (CS) as the goal gene and used it with the most and least stable reference genes, and various combinations of them, to observe the relatively expressions levels under various treatments. The expression and synthesis pathways under different treatments and the expression patterns of 14 genes under different experimental treatments were analyzed. The Ct values were determined based on the 2^−ΔΔCT^ methodology using the raw data. Three independent repetitions were performed on each sample. Furthermore, correlation graphs were plotted using GraphPad Prism 9 software, TBtools (2.012) [56], Adobe illustrator 2021, and SPSS 17.0 software.

## 5. Conclusions

In this research, using tissue culture seedling leaves, 12 reference genes were separated to ensure the stability of RT-qPCR normalization. According to the analysis, *CDC6* and *NCBP2* were the best reference genes to normalize the expression levels under different treatments. Using *CDC6* and *NCBP2* as reference genes, RT-qPCR was performed on the genes for the mogrosides synthesis of *S. siamensis*, and the expression patterns of the key genes were analyzed under various treatments to help identify genes important for mogrosides synthesis.

## Figures and Tables

**Figure 1 plants-13-02449-f001:**
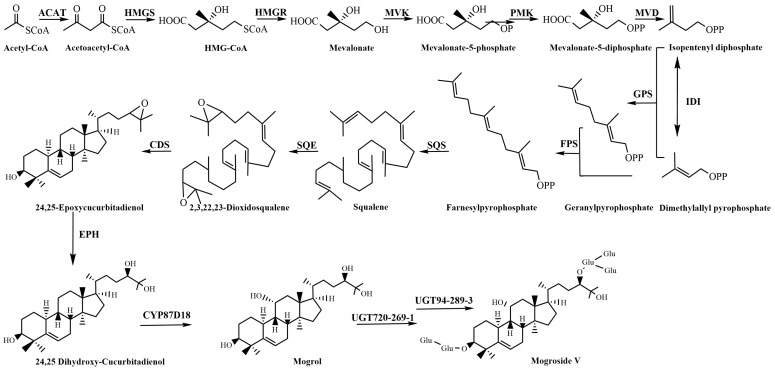
Schematic diagram of the synthesis pathway of mogrosides.

**Figure 2 plants-13-02449-f002:**
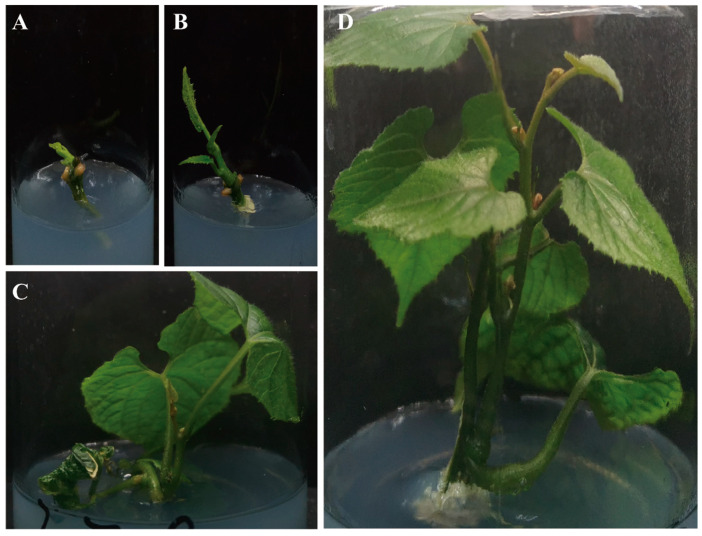
Growth condition of *S. siamensis* at different periods: (**A**) exosome bud point (botany); (**B**) 2 weeks of exosome bud site growth; (**C**) 1 month of exosome bud site growth; (**D**) 3 months of exosome bud site growth.

**Figure 3 plants-13-02449-f003:**
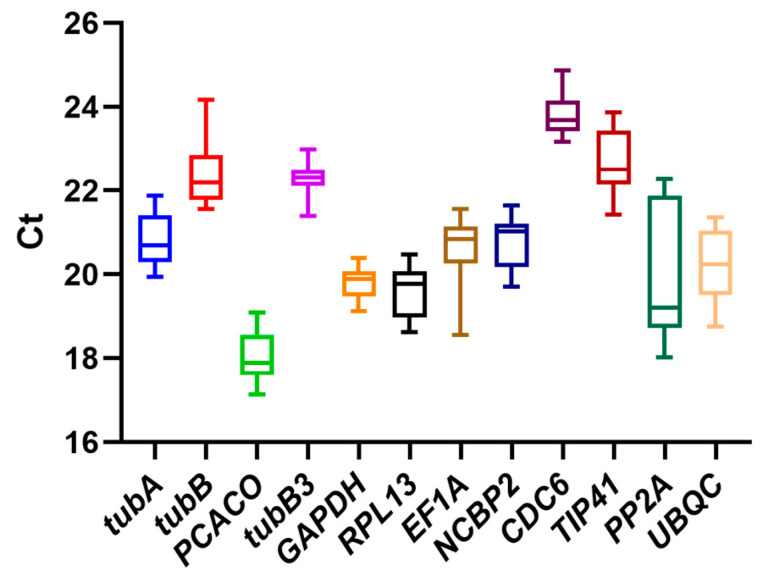
Box plots of the distributions of Ct values for the twelve reference genes. Lines shown in the box-plot graph of Ct values display the median values. Lower and upper boxes represent the 25th percentile to the 75th percentile. Whiskers indicate the maximum and minimum values.

**Figure 4 plants-13-02449-f004:**
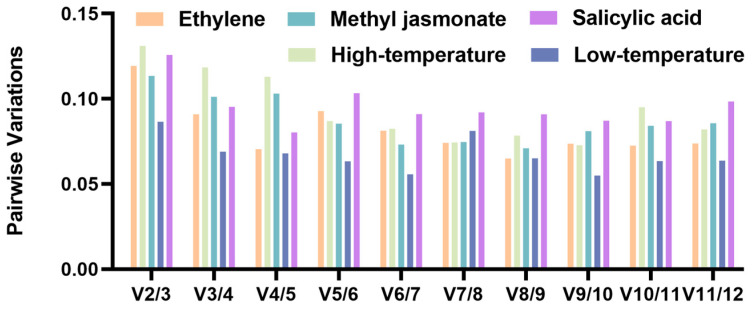
Pairwise variance (Vn/n + 1) fraction of twelve references measuring with GeNorm.

**Figure 5 plants-13-02449-f005:**
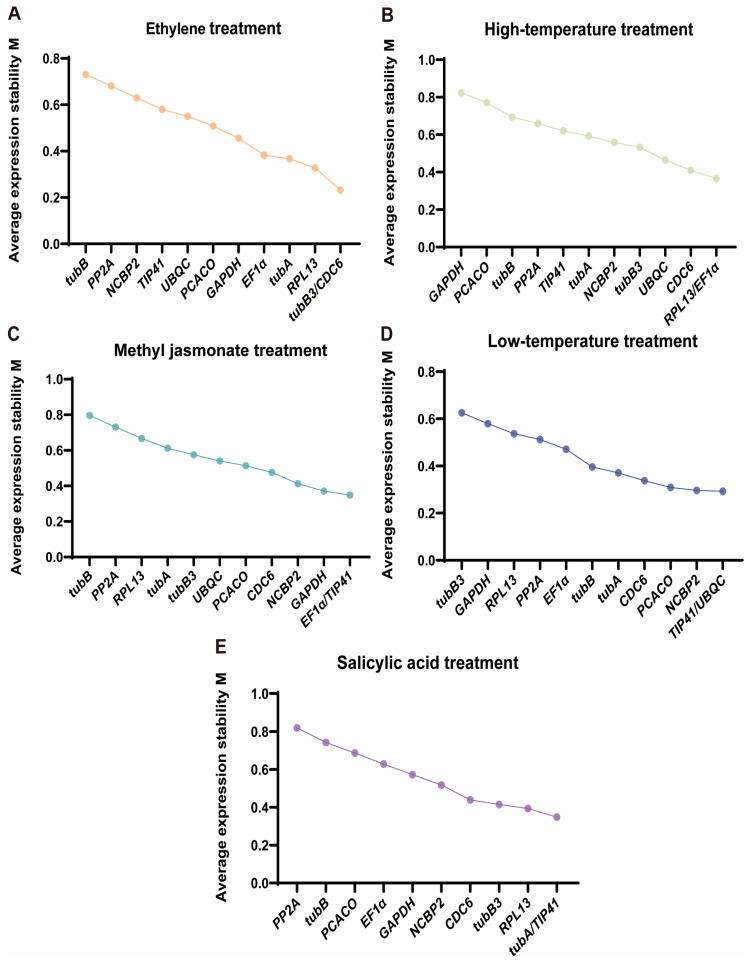
Average expression stability values (M) of ten candidate reference genes using GeNorm. (**A**) M-value sorting under EtH treatment; (**B**) M-value sorting under MeJA treatment; (**C**) M-value sorting under Low Tem treatment; (**D**) M-value sorting under High Tem treatment; (**E**) M-value sorting under SA treatment.

**Figure 6 plants-13-02449-f006:**
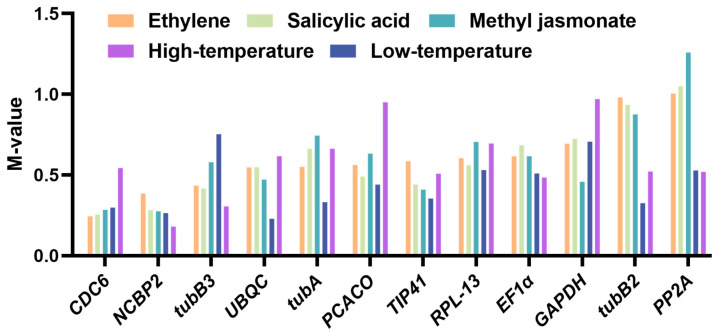
Results of analyzing twelve reference genes using NormFinder.

**Figure 7 plants-13-02449-f007:**
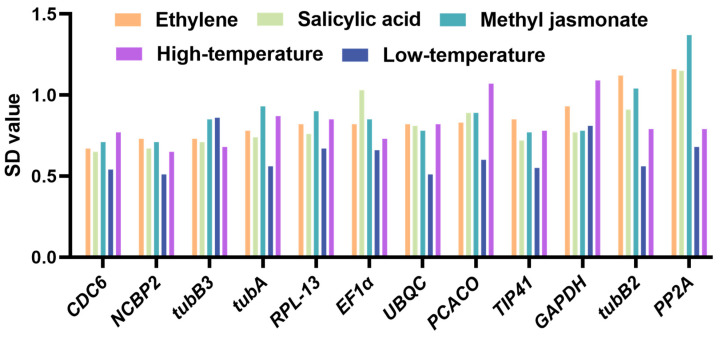
Results of analyzing twelve reference genes using Delta Ct.

**Figure 8 plants-13-02449-f008:**
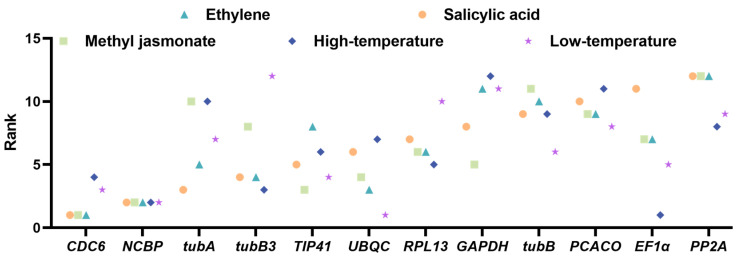
Results of sequencing twelve reference genes using RefFinder.

**Figure 9 plants-13-02449-f009:**
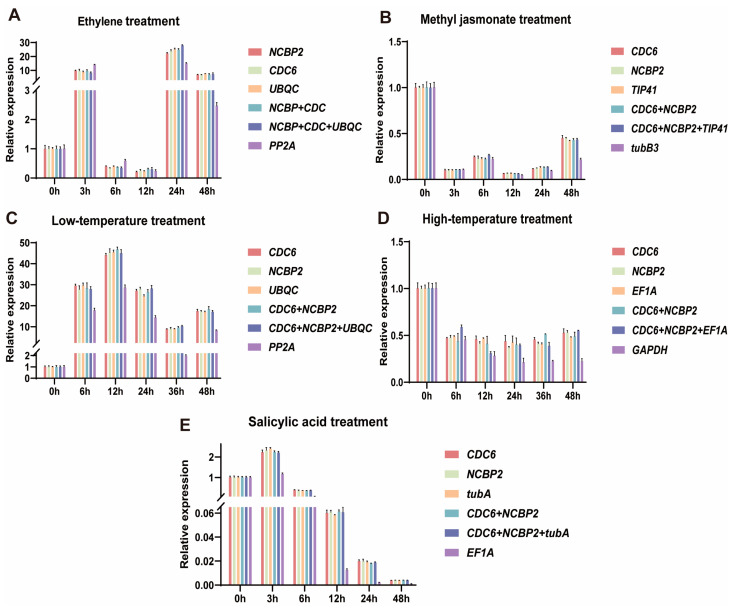
Expression patterns of *SsCS* genes under different treatments. Validated by the relative expression levels of the SsCS gene. (**A**) Expression pattern results of EtH treatment for 0, 3, 6, 12, 24, and 48 h; (**B**) expression pattern results of MeJA treatment for 0, 3, 6, 12, 24 and 48 h; (**C**) expression pattern results of low-temperature treatment for 0, 6, 12, 24, 36, and 48 h; (**D**) expression pattern results of high-temperature treatment for 0, 6, 12, 24, 36, and 48 h; (**E**) expression pattern results of SA treatment for 0, 3, 6, 12, 24, and 48 h.

**Figure 10 plants-13-02449-f010:**
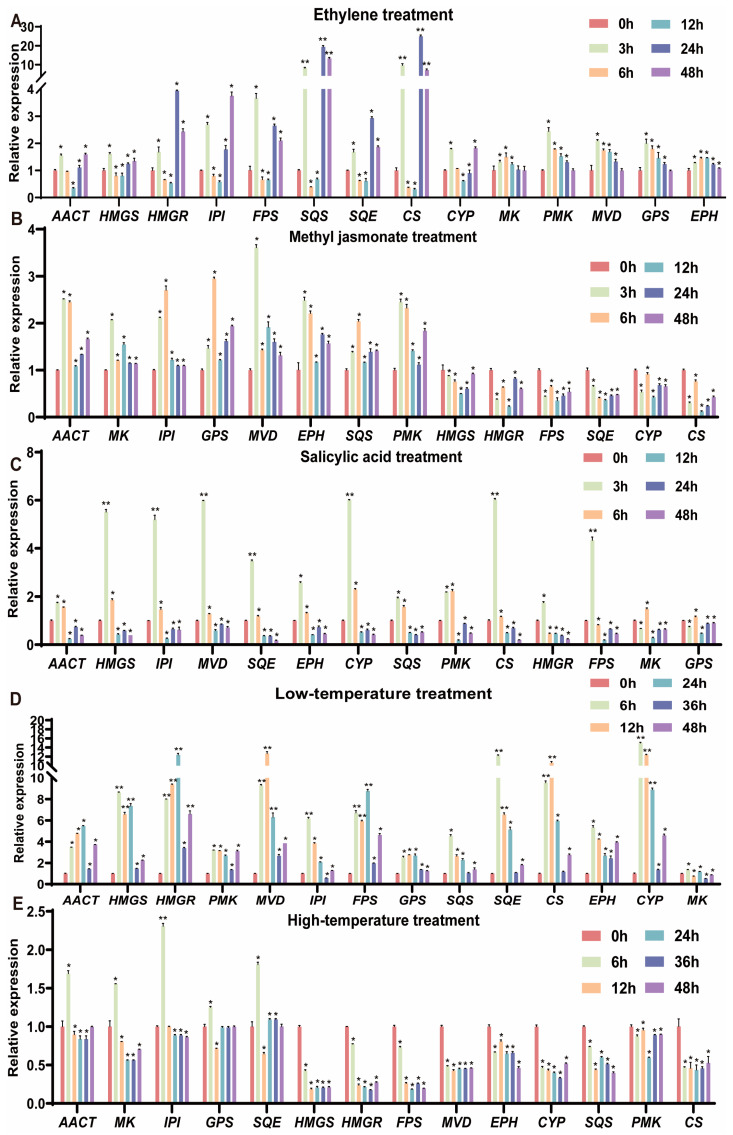
Gene expression patterns of mogrosides synthesis pathway under different treatments. Statistical analysis was performed using an independent samples *t*-test, with six time periods according to different treatments. The reference genes used were a combination of CDC6 and NCBP2 to obtain the CT values. Each sample underwent three parallel experiments. (**A**) Expression pattern calculated by RT-qPCR in EtH treatment for 0, 3, 6, 12, 24, and 48 h. (**B**) Expression pattern calculated by RT-qPCR in MeJA treatment for 0, 3, 6, 12, 24, and 48 h. (**C**) Expression pattern calculated by RT-qPCR in SA treatment for 0, 3, 6, 12, 24, and 48 h. (**D**) Expression pattern calculated by RT-qPCR in low-temperature treatment for 0, 6, 12, 24, 36, and 48 h. (**E**) Expression pattern calculated by RT-qPCR in high-temperature treatment for 0, 6, 12, 24, 36, and 48 h. (** indicates *p* < 0.01, * indicates *p* < 0.05.).

**Table 1 plants-13-02449-t001:** Gene information, primer amplification efficiency, and *R*^2^ values.

Gene	Gene ID	PCR Efficiency (%)	*R* ^2^	Gene	E.C	Gene ID	PCR Efficiency (%)	*R* ^2^
*SsRPL-13*	*Ribosomal protein L-13*	102.6	0.995	*SsAACT*	2.3.1.9	*acetyl-CoA C-acetyltransferase*	101.1	0.993
*SsCDC6*	*Cell division control protein 6*	100.5	0.994	*SsHMGS*	2.3.3.10	*hydroxymethylglutaryl-CoA synthase*	98.7	0.996
*SsTIP41*	*TIP41-like family protein*	100.3	0.995	*SsHMGR*	1.1.1.34	*hydroxymethylglutaryl-CoA reductase*	97.8	0.998
*SstubB*	*β-tubulin2*	100.6	0.993	*SsMK*	2.7.1.36	*mevalonate kinase*	99.5	0.995
*SsGAPDH*	*glyceraldehyde-3-phosphate dehydrogenase*	98.8	0.994	*SsPMK*	2.7.4.2	*phosphomevalonate kinase*	100.2	0.996
*SstubA*	*α-tubulin2*	101.3	0.996	*SsMVD*	4.1.1.33	*diphosphomevalonate decarboxylase*	100.4	0.994
*SsEF1α*	*Elongation factor 1α*	99.7	0.993	*SsIPI*	5.3.3.2	*Isopentenyl-diphosphate Delta-isomerase*	99.7	0.996
*SsNCBP2*	*Nuclear cap-binding protein subunit 2*	102.2	0.997	*SsGPS*	2.5.1.10	*geranylpyrophosphate synthetase*	102.1	0.997
*SsUBQC*	*ubiquitin C*	101.3	0.995	*SsFPS*	2.5.1.21	*farnesyl pyrophosphate synthetase*	99.5	0.998
*SstubB3*	*β-tubulin3*	99.6	0.996	*SsSQS*	1.14.14.17	*squalene synthase*	101.1	0.994
*SsPP2A*	*Protein phosphatase 2A*	100.4	0.994	*SsSQE*	1.14.19.-	*squalene epoxidase*	100.6	0.996
*SsPcACO*	*1-aminocyclopropane-1-carboxylate oxidase A*	100.2	0.998	*SsCS*	5.4.99.33	*Cucurbitadienol synthase*	99.4	0.998
*SsEPH*	*Epoxide hydrolase*	98.9	0.994	*SsCYP*	1.14.-. -	*Cytochrome P450*	99.7	0.995

**Table 2 plants-13-02449-t002:** Stability of expression (CV) ± (SD) of twelve reference genes using BestKeeper.

	Low-Temperature Treatment			High-Temperature Treatment			Ethylene Treatment			Salicylic Acid Treatment			Methyl Jasmonate Treatment		
Rank	Std Dev	Cv%	Gene Name	Std Dev	Cv%	Gene Name	Std Dev	Cv%	Gene Name	Std Dev	Cv%	Gene Name	Std Dev	Cv%	Gene Name
1	0.26	1.46	*EF1α*	0.3	1.48	*EF1α*	0.33	1.6	*PP2A*	0.35	1.57	*CDC6*	0.34	1.65	*UBQC*
2	0.36	1.48	*CDC6*	0.32	1.49	*NCBP2*	0.39	1.73	*CDC6*	0.37	1.63	*TIP41*	0.45	2.02	*tubB2*
3	0.37	1.56	*tubB2*	0.37	1.58	*CDC6*	0.41	1.84	*TIP41*	0.39	2.13	*PCACO*	0.44	2.11	*NCBP2*
4	0.33	1.56	*PP2A*	0.49	2.2	*tubB3*	0.52	2.32	*tubB2*	0.43	2.28	*GAPDH*	0.54	2.37	*TIP41*
5	0.32	1.67	*UBQC*	0.52	2.34	*PP2A*	0.5	2.58	*NCBP2*	0.43	2.28	*tubB3*	0.54	2.41	*CDC6*
6	0.37	2.02	*RPL-13*	0.55	2.41	*TIP41*	0.54	2.58	*tubB3*	0.5	2.43	*NCBP2*	0.6	2.82	*tubB3*
7	0.47	2.14	*TIP41*	0.51	2.66	*UBQC*	0.64	3.08	*GAPDH*	0.66	3.07	*tubB2*	0.73	3.74	*tubA*
8	0.46	2.24	*NCBP2*	0.5	2.67	*RPL-13*	0.59	3.13	*tubA*	0.69	3.27	*UBQC*	0.75	3.84	*GAPDH*
9	0.53	2.47	*tubA*	0.68	2.92	*tubB2*	0.67	3.8	*PCACO*	0.64	3.34	*EF1α*	0.74	3.97	*RPL-13*
10	0.57	3.22	*PCACO*	0.82	3.77	*GAPDH*	0.75	4.2	*RPL-13*	0.8	4.18	*tubA*	0.79	4.37	*PCACO*
11	0.69	3.25	*tubB3*	0.73	3.81	*tubA*	0.84	4.36	*EF1α*	0.92	4.58	*PP2A*	0.81	4.7	*PP2A*
12	0.88	4.63	*GAPDH*	1.05	5.37	*PCACO*	0.97	4.84	*UBQC*	0.87	4.78	*RPL-13*	0.99	5.08	*EF1α*

**Table 3 plants-13-02449-t003:** Different combinations of plant hormones.

Level	Factor		
	NAA (mg/L)	IBA (mg/L)	Activated Carbon (mg/L)
1	0.1	0.3	100
2	0.3	0.5	200
3	0.5	0.7	300

## Data Availability

The *S. siamensis* genome database was deposited in the National Center for Biotechnology Information (https://dataview.ncbi.nlm.nih.gov/object/SRR22947134?reviewer=qmfs7mc075pohiv011jqjceh94, accessed on 2 January 2023).

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
