# Peer review of "Selection of Reference Genes in Siraitia siamensis and Expression Patterns of Genes Involved in Mogrosides Biosynthesis"

_plants, 2024, doi:10.3390/plants13172449_

Round 1

Reviewer 1 Report

Comments and Suggestions for Authors

This article describes that the selection of genes responsible to productivity of mogroside V, fairy expensive triterpene glycoside, in S. siamensis based on the works which each authors have carried out previously. The structural genes  for key enzymes in biosynthesis of mogroside V were studied by the authors. 

 The experimentals look sound and cultural conditions are quite neat! The referee feels , squalene epoxidase(SQE) and cucurubitadienol synthase(CS) are well defined, but as for the P450 enzyme which should oxidize 11 position of the triterpene was not clearly written in the manuscript. Because there are so many CYPs in any plants, so that the CYP450 should be defined specially for this biosynthetic process.  And also the referee feels why glycosyl transferases in this plant are not listed in the tables which were already known. 

Small points, in Fig.10 why the order of genes in X-axis are changes by every graphs?  Etc, Tim etc had better spell out as ethylene or temperature etc.

Author Response

Response to Reviewer 1 Comments

Point 1: The referee feels, squalene epoxidase (SQE) and cucurubitadienol synthase (CS) are well defined, but as for the P450 enzyme which should oxidize 11 position of the triterpene was not clearly written in the manuscript. Because there are so many CYPs in any plants, so that the CYP450 should be defined specially for this biosynthetic process. And, the referee feels why glycosyl transferases in this plant are not listed in the tables which were already known.

Response 1: Thank you for your valuable feedback on this paper. We have reinstated the definition of CYP450 enzyme in the main text as follow:

“Cytochrome P450 monooxygenase (CYP450) is responsible for the hydroxylation at C3, C11, C24, and C25 required to produce mogrol in the mogroside biosynthesis pathway. Among these, the CYP87D18 enzyme is a multifunctional enzyme that oxidizes the C-11 position of cucurbitadienol to form mogrol and 11-O-mogrol [22].”

Regarding why glycosyltransferases were not focused on in mogroside synthesisthe pathway, it is because there was no expression of UGT in the leaves of S. siamensis seedlings [1], this gene is expressed during the fruit ripening stage, and corresponding values were not found in RT-qPCR. Therefore, related data was not listed in the table.

  1. Cui, S.; Zang, Y.; Xie, L.; Mo, C.; Su, J.; Jia, X.; Luo, Z.; Ma, X. Post-Ripening and Key Glycosyltransferase Catalysis to Promote Sweet Mogrosides Accumulation of Siraitia Grosvenorii Fruits. Molecules2023, 28, 4697, doi:10.3390/molecules28124697.

Point 2: Small points, in Fig.10 why the order of genes in X-axis are changes by every graphs?  Etc, Tim etc had better spell out as ethylene or temperature etc.

Response 2: Thank you for your suggestions. We have corrected all incorrect abbreviations in the pictures to their full better spellings. In Fig. 10, the order of genes showed with consistent expression trends together. The gene expression trends vary under different treatments, resulting in different gene orders in X-axis.

Reviewer 2 Report

Comments and Suggestions for Authors

In this correspondence, authors first found available reference genes in Siraitia siamensis by using five different algorithms and further demonstrated their finding in the mogrosides biosynthesis pathway, which would be very helpful for the following research on the biosynthesis pathway of secondary metabolites in Siraitia siamensis. However, this work should be accepted after minor revision. The points are listed below.

Major Points:

1.    The abstract should be further simplified and rephrased to avoid being conversational.

2.    I recommend that all the primer sequences be put in the supplementary information instead of in the manuscript. Authors may find another way to show PCR efficiency and R2.(Table 1-3)

3.    Please pay attention to the figure legends. There is no Table 3 in the manuscript, while there is a Table 4.

4.    For Figure 3, did all the data come from three repeat experiments? All the data came from the same sample (plant tissues)? If not, making such a figure is confusing because the error bar seems very dramatic.

Comments on the Quality of English Language

The abstract should be further simplified and rephrased to avoid being conversational.

Author Response

Response to Reviewer 2 Comments

Point 1: The abstract should be further simplified and rephrased to avoid being conversational.

Response 1: Thank you for your suggestions on abstract writing. Sentences emphasizing scientific value have been added to the abstract, as follows:

Siraitia siamensis is a traditional Chinese medicinal herb. In this study, using S. siamensis cultivated in vitro, twelve candidate reference genes under various treatments were analyzed for their expression stability by using algorithms such as GeNorm, NormFinder, BestKeeper, Delta CT, and RefFinder. The selected reference genes were then used to characterize the gene expression of cucurbitadienol synthase, which is a rate-limiting enzyme for mogroside biosynthesis. The results showed that CDC6 and NCBP2 expression was the most stable across all treatments and are the best reference genes under the tested conditions. Utilizing the validated reference genes, we analyzed the expression profiles of genes related to the synthesis pathway of mogroside in S. siamensis in response to a range of abiotic stresses. The findings of this study provide clear standards for gene expression normalization in Siraitia plants and exploring the rationale behind differential gene expression related to mogroside synthesis pathways.”

Point 2: I recommend that all the primer sequences be put in the supplementary information instead of in the manuscript. Authors may find another way to show PCR efficiency and R2.(Table 1-3)

Response 2: Thank you for your suggestions. For PCR amplification efficiency and R2, we have made the following modifications in the main text to reflect this, and the primer sequence information is placed in the supplementary information (Table S1 and S6).

Table 1. Gene information, primer amplification efficiency, and R2 values.

Gene

Gene ID

PCR efficiency (%)

R2

Gene

E.C

Gene ID

PCR efficiency (%)

R2

SsRPL-13

Ribosomal protein L-13

102.6

0.995

SsAACT

2.3.1.9

acetyl-CoA C-acetyltransferase

101.1

0.993

SsCDC6

Cell division control protein 6

100.5

0.994

SsHMGS

2.3.3.10

hydroxymethylglutaryl-CoA synthase

98.7

0.996

SsTIP41

TIP41-like family protein

100.3

0.995

SsHMGR

1.1.1.34

hydroxymethylglutaryl-CoA reductase

97.8

0.998

SstubB

β-tubulin2

100.6

0.993

SsMK

2.7.1.36

mevalonate kinase

99.5

0.995

SsGAPDH

glyceraldehyde-3-phosphate dehydrogenase

98.8

0.994

SsPMK

2.7.4.2

phosphomevalonate kinase

100.2

0.996

SstubA

α-tubulin2

101.3

0.996

SsMVD

4.1.1.33

diphosphomevalonate decarboxylase

100.4

0.994

SsEF1α

Elongation factor 1α

99.7

0.993

SsIPI

5.3.3.2

Isopentenyl-diphosphate Delta-isomerase

99.7

0.996

SsNCBP2

Nuclear cap-binding protein subunit 2

102.2

0.997

SsGPS

2.5.1.10

geranylpyrophosphate synthetase

102.1

0.997

SsUBQC

ubiquitin C

101.3

0.995

SsFPS

2.5.1.21

farnesyl pyrophosphate synthetase

99.5

0.998

SstubB3

β-tubulin3

99.6

0.996

SsSQS

1.14.14.17

squalene synthase

101.1

0.994

SsPP2A

Protein phosphatase 2A

100.4

0.994

SsSQE

1.14.19.-

squalene epoxidase

100.6

0.996

SsPcACO

1-aminocyclopropane-1-carboxylate oxidase A

100.2

0.998

SsCS

5.4.99.33

Cucurbitadienol synthase

99.4

0.998

SsEPH

Epoxide hydrolase

98.9

0.994

SsCYP

Cytochrome P450

99.7

0.995

Point 3: Please pay attention to the figure legends. There is no Table 3 in the manuscript, while there is a Table 4.

Response 3: I'm sorry for the mistake made during the writing process of the paper. We have corrected the erroneous legend numbers.

Point 4: For Figure 3, did all the data come from three repeat experiments? All the data came from the same sample (plant tissues)? If not, it is quite confusing to make such a figure because the error bar seems very dramatic.

Thank you for your suggestions for Figure 3. All RT-qPCR experimental data in Figure 3 were obtained from three independent experimental replicates, and the data do not originate from a single experimental sample but from samples at 0 hour under different treatments. We will provide detailed explanations and make modifications in the methods and figure captions. The specific changes are as follows:

In Figure 3 legend “Lines shown in the box-plot graph of Ct value display the median values. Lower and upper boxes represent the 25th percentile to the 75th percentile. Whiskers indicate the maximum and minimum values.”

In method “First different treatments 0 hour of template cDNA (~100 ng) was used for all samples. box plots of the cycle threshold (Ct) distribution of the reference genes in the control group were plotted using GraphPad Prism 9 software. Three replicates were used for sample tested. ”

Reviewer 3 Report

Comments and Suggestions for Authors

The research in the paper is devoted to evaluation of the few S. siamensis genes for identification of the best of those for application as reference genes for RT-PCR analysis in this plant. In this work authors also evaluate expression levels of the genes of Siraitia specialized metabolites biosynthesis in response to different treatments. These genes were for the first time tested in Siraitia under conditions applied. Emphasizing it and focusing more on these data will bring higer significance to the work. 

The paper needs some improvements:

Introduction should be corrected as some statement are not supported by the relevant references. Please, check it out throughout the paper. For example, line 45 refers to [4], which is devoted to the role of Polygonatum WRKY TFs in plant stress resistance, but not to potential of Siraitia extracts in application as sweetener. [5] is devoted to Siraitia chloroplast genomes sequencing and analysis, but not to effect of mogroside on human physiology [lines 47-48]. [6] does not contain any information on sweetness of siamenoside I, as it is stated in lines 47-49

[3] mentions S. grosvenorii, but not S. siamensis to be used in traditional medicine for cold treatment.

The figure legends throughout the paper and text in results section can by improved to bring better understanding of what we see on figures. For example:

Lines 307-314 What is best and what is worse? Even though, the conclusions on more and less stable reference genes were done above, the description of results, demonstrated in figure requires clarification.

Figure 10- What was use as a reference gene for the data in Figure? Please, clarify it in legend. The general scheme of experiment, demonstrated in figures should be clear from the relevant results section and/or from figure legends (it is relevant for Fig.9 as well) – how the treatment was applied and the hours marked should be explained.   

Lines 232, 268 “sequencing”? What type of machine was used for RT-PCR?

It is not clear what was used as a sample for all the experiments. “ Organism” in  line 490 – is it the whole plant, with/without roots? 

Other issues:

In line 24, please, replace “vitro cultures of S. siamensis’ with “S. siamensis cultivated in vitro” or something like this. “Vitro cultures of S. siamensis “ can be understood by reader like cell culture.

Lines 117-118 what are those reports? Which genes were suggested as reference for S. siamensis there?

Author Response

Response to Reviewer 3 Comments

Point 1: Introduction should be corrected as some statement are not supported by the relevant references. Please, check it out throughout the paper. For example, line 45 refers to [4], which is devoted to the role of Polygonatum WRKY TFs in plant stress resistance, but not to potential of Siraitia extracts in application as sweetener. [5] is devoted to Siraitia chloroplast genomes sequencing and analysis, but not to effect of mogroside on human physiology [lines 47-48]. [6] does not contain any information on sweetness of siamenoside I, as it is stated in lines 47-49; [3] mentions S. grosvenorii, but not S. siamensis to be used in traditional medicine for cold treatment.

Response 1: Apologies for any errors in the citation of the literature. The latest literature has been added to the introduction section and incorrect references have been removed. The specific modifications are as follows:

“It has been used for many years by traditional Chinese medicine for the treatment of congested lungs, common colds and laryngitis [2,3]. Siraitia species include Siraitia grosvenorii and S. siamensis, whose main bioactive compounds are mogrosides, a type of triterpenoid sweetener. The fruit extracts can be used as sugar-free health food and beverage supplements and sweeteners [4] because of the presence of naturally low-calorie mogroside V. It can be used as a sugar substitute for people with diabetes, as it does not raise blood sugar levels [5]. S. siamensis is rich in siamenoside I, being about 560 times as sweet as sucrose with approximately 1.4 times as sweet as aspartame [6]. “

  1. 2. Tang, Q.; Ma, X.; Mo, C.; Wilson, I.W.; Song, C.; Zhao, H.; Yang, Y.; Fu, W.; Qiu, D. An Efficient Approach to Finding Siraitia grosvenoriiTriterpene Biosynthetic Genes by RNA-Seq and Digital Gene Expression Analysis. BMC Genomics2011, 12, 343, doi:10.1186/1471-2164-12-343.
  2. 3. Kasai, R.; Nie, R.-L.; Nashi, K.; Ohtani, K.; Zhou, J.; Tag, G.-D.; Tanaka, O. Sweet Cucurbitane Glycosides from Fruits of Siraitia siamensis(Chi-Zi Luo-Han-Guo), a Chinese Folk Medicine. Agricultural and Biological Chemistry1989, 53, 3347–3349, doi:10.1271/bbb1961.53.3347.
  3. 4. Pawar, R.S.; Krynitsky, A.J.; Rader, J.I. Sweeteners from Plants--with Emphasis on Stevia rebaudiana(Bertoni) and Siraitia grosvenorii (Swingle). Anal Bioanal Chem2013, 405, 4397–4407, doi:10.1007/s00216-012-6693-0.
  4. 5. Òªiçek, S.S. Structure-Dependent Activity of Plant-Derived Sweeteners. Molecules2020, 25, 1946, doi:10.3390/molecules25081946.
  5. 6. Xu, Y.; Zhao, L.; Chen, L.; Du, Y.; Lu, Y.; Luo, C.; Chen, Y.; Wu, X. Selective Enzymatic α-1,6- Monoglucosylation of Mogroside IIIE for the Bio-Creation of α-Siamenoside I, a Potential High-Intensity Sweetener. Food Chemistry2021, 359, 129938, doi:10.1016/j.foodchem.2021.129938.

Point 2: The figure legends throughout the paper and text in results section can by improved to bring better understanding of what we see on figures. For example:

Lines 307-314 What is best and what is worse? Even though, the conclusions on more and less stable reference genes were done above, the description of results, requires clarification.

Response 2: Thank you for your valuable comments. It should not be described as the best and worst genes; instead, it should be characterized as the most stable and less stable reference genes. We will strengthen the description of the results and legends demonstrated in Fig. 9 from lines 307 to 314:

In result: “As shown in Fig. 9, when the three most stable genes were used individually or in combination as reference genes for normalization, the relative expression pattern of SsCS showed similar trends. However, when an unstable gene was used for relative quantification, the relative expression levels of SsCS showed significant fluctuations. For example, under Eth treatment from 0-48 h, when stable genes (NCBP2, CDC6, UBQC and their combinations) were used as reference genes, the expression level of SsCS was highest at 24h and lowest at 12h, with an overall trend of first decreasing then increasing. In contrast, when the least stable gene (PP2A) was used, the expression level of SsCS was high at 3h and 24h, with a different overall trend (Fig. 9A). It is evident that using unstable reference genes for gene expression analysis in S. siamensis can lead to unreliable results. Additionally, in the SA, MeJA, high temperature, and low temperature treatment groups, CDC6 and NCBP2 were stable reference genes, and the expression levels and trends of SsCS under different groups pointed to similar conclusions (Fig. 9BCDE).”

In legends: “Validated by the relative expression levels of the SsCS gene. A: Expression pattern results of EtH treatment about 0, 3, 6, 12, 24 and 48 h; B: Expression pattern results of MeJA treatment about 0, 3, 6, 12, 24 and 48 h; C: Expression pattern results of Low-treatment treatment about 0, 6, 12, 24, 36 and 48 h; D: Expression pattern results of High-treatment treatment about 0, 6, 12, 24, 36 and 48 h; E: Expression pattern results of SA treatment about 0, 3, 6, 12, 24 and 48 h.”

Point 3: Figure 10- What was use as a reference gene for the data in Figure? Please, clarify it in legend. The general scheme of experiment, demonstrated in figures should be clear from the relevant results section and/or from figure legends (it is relevant for Fig.9 as well) – how the treatment was applied and the hours marked should be explained.  

Response 3: Thank you for your valuable comments. CDC6 and NCBP2 were used as the reference gene to identify and validate mogrosides synthesis pathways genes. We will strengthen the description of the results and legends demonstrated in Fig. 10.

In result: CDC6 and NCBP2 were used as the reference gene to identify and validate mogrosides synthesis pathways genes whose expression alters with the application of MeJA, EtH, SA, heat and cold stress. (Table 1 and Table S6) RT-qPCR analysis was performed on leaves from tissue culture seedlings treated with MeJA, SA, and EtH at 0, 3, 6, 12, 24, 48 h, as well as on leaves subjected to high and low temperature treatments at 0, 6, 12, 24, 36, and 48 h.”

In legends: “The CDC6 and NCBP2 genes were used to analyze the expression patterns of genes related to the mogroside biosynthesis pathway. A: Expression pattern calculated by RT-qPCR in EtH treatment about 0, 3, 6, 12, 24 and 48 h. B: Expression pattern calculated by RT-qPCR in MeJA treatment about 0, 3, 6, 12, 24 and 48 h. C: Expression pattern calculated by RT-qPCR in SA treatment about 0, 3, 6, 12, 24 and 48 h. D: Expression pattern calculated by RT-qPCR in Low-temperature treatment about 0, 6, 12, 24, 36 and 48 h. E: Expression pattern calculated by RT-qPCR in High-temperature treatment about 0, 6, 12, 24, 36 and 48 h. (** indicates p < 0.01, * indicates p < 0.05.)”

Point 4: Lines 232, 268 “sequencing”? What type of machine was used for RT-PCR?

Response 4: I'm sorry for the mistake made in these two instances for the paper. We should have referred to analyzing the results of different algorithms, not sequencing the results. Corrections will be made. The experiments we conducted utilized the ABI7300 real-time PCR instrument (ABI7300, Applied Biosystems, USA) for RT-qPCR analysis.

Point 5: It is not clear what was used as a sample for all the experiments. “ Organism” in  line 490 – is it the whole plant, with/without roots?

Response 5: I'm sorry for the mistake made in the sample for all the experiments. We used the leaf section of S. siamensis as our experimental sample and made modifications in the text, as follow: “Rapidly freeze 50-100 mg of S. siamensis leaf samples in liquid nitrogen and then grind to a fine powder using a mortar and pestle.”

Point 6: In line 24, please, replace “vitro cultures of S. siamensis’ with “S. siamensis cultivated in vitro” or something like this. “Vitro cultures of S. siamensis “ can be understood by reader like cell culture.

Response 6: Thank you for your suggestions on abstract writing, as follows:

“In this study, using S. siamensis cultivated in vitro, twelve candidate reference genes under various treatments were analyzed for their expression stability by using algorithms such as GeNorm, NormFinder, BestKeeper, Delta CT, and RefFinder.

Point 7: Lines 117-118 what are those reports? Which genes were suggested as reference for S. siamensis there?

Response 7: Thank you for your suggestions. Currently, there are reports on the reference genes of S. grosvenorii, but there are no reports on S. siamensis. The expression in the text is incorrect, and we have modified it, as follow:

“Currently, there are no reports on the selection and validation of reference genes for S. siamensis. To ensure reliable RT-qPCR studies in S. siamensis, it is necessary to identify reference genes.”

Round 2

Reviewer 3 Report

Comments and Suggestions for Authors

Something is wrong with line 346 in figure legend. Please, correct it. Figure legend 9 – “treatment for” instead of “treatment about”?

Fig.10 is not mentioned in the text ( only 10A)

Figure legend 10 – What type of statistical analysis is applied, what is the sample size (n)?

It is still not clear which of the genes (CDC6 or NCBP2) was used to obtain the figure? These certain values were obtained by calculation vs one of them – which one? If they both gave similar results, it should be clarified ( maybe to place the data for another into supplement)

The supplementary data file and the references to that should be checked out thoughout the paper:

Line 359 - There is no table S6 in the supplementary data file

Lines 591, 592 – Tables, but not figures

Line 529 Table S1 does not contain the data on primers, Table S6 is missing

On my opinion it is reasonable to emphasize in introduction or maybe even abstract that the protocol for in vitro cultivation and propagation of S. siamensis was developed for the first time. 

Author Response

Response to Reviewer 1 Comments

Point 1: Something is wrong with line 346 in figure legend. Please, correct it.

Figure legend 9 – “treatment for” instead of “treatment about”?

Fig.10 is not mentioned in the text ( only 10A)

Response 1: I'm sorry for the mistake made during the writing process of the paper and have reviewed and corrected the corresponding sections.

Point 2: Figure legend 10 – What type of statistical analysis is applied, what is the sample size (n)?

Response 2: Thank you for your suggestions. Figure 10 utilized the -2ΔΔCT method for data calculation and SPSS for independent samples T-test statistical analysis. This statistical method was performed according to different treatments, with 6 time periods per treatment and a total of 14 candidate synthetic pathway genes. The reference genes used were a combination of CDC6 and NCBP2. Each sample underwent three parallel experiments 

In legend: “Statistical analysis was performed using an independent samples T-test, with six time periods according to different treatments. The reference genes used were a combination of CDC6 and NCBP2 to obtain the CT values. Each sample underwent three parallel experiments.”

Point 3: It is still not clear which of the genes (CDC6 or NCBP2) was used to obtain the figure? These certain values were obtained by calculation vs one of them – which one? If they both gave similar results, it should be clarified ( maybe to place the data for another into supplement)

Response 3: After preliminary computational statistics in this experiment, it was found that the two reference genes CDC6 and NCBP2 possess good stability. Similarly, the optimal number of reference genes was determined to be 2 through Genorm variation coefficient calculation. Therefore, the gene expression of the synthetic pathway after different treatments was calculated based on a combination of the two reference genes CDC6 and NCBP2, not on a single reference gene.

Point 4: The supplementary data file and the references to that should be checked out though-out the paper:

Line 359 - There is no table S6 in the supplementary data file

Lines 591, 592 – Tables, but not figures

Line 529 Table S1 does not contain the data on primers, Table S6 is missing

Response 4: I'm sorry for the mistake made during the writing process of the paper and have reviewed and corrected the entire manuscript, and added the corresponding supplementary data in the supplementary materials.

Point 5: On my opinion it is reasonable to emphasize in introduction or maybe even abstract that the protocol for in vitro cultivation and propagation of S. siamensis was developed for the first time.

Response 4: Thank you for the reviewer's valuable comments. We have strengthened the description of the in vitro tissue culture of S. siamensis in the introduction:

Preface: "Currently, there are no reports on the in vitro tissue culture of S. siamensis; therefore, the development of an in vitro culture and propagation protocol for this species is urgently needed for research purposes. By analyzing and modifying methods of in vitro tissue culture for S. grosvenorii, results were obtained through orthogonal experiments."